# Are fast scramblers good thermal baths?

A. Larzul[1], S. J. Thomson[2,1], M. Schirò[1],

**1** JEIP, USR 3573 CNRS, Collège de France, PSL Research University, 11 Place Marcelin
Berthelot, 75321 Paris Cedex 05, France
**2** Dahlem Centre for Complex Quantum Systems, Freie Universität, 14195 Berlin, Germany

April 14, 2022

## Abstract

The Sachdev-Ye-Kitaev ($SYK_4$) model has attracted attention for its fast scrambling properties and its thermalization rate that is set only by the temperature. In this work we ask the question of whether the $SYK_4$ model is also a good thermal bath, in the sense that it allows a system coupled to it to thermalize. We address this question by considering the dynamics of a system of $N$ random non-interacting Majorana fermions coupled to an $SYK_4$ bath with $M$ Majorana fermions that we solve with Keldysh techniques in the limit of $M \gg N \gg 1$. We compare this nonequilibrium setting with a conventional bath made of free non-interacting degrees of freedom with a continous spectrum. We show that the $SYK_4$ bath is more efficient in thermalising the system at weak coupling, due to its enhanced density of states at low frequency, while at strong system-bath couplings both type of environments give rise to a similar time scale for thermalisation.

## 1   Introduction

Recent years have seen a resurgence of interest around questions at the foundation of quantum statistical mechanics, concerning thermalization, chaos and ergodicity in isolated quantum many-body systems far from equilibrium [1,2]. This renewed interest has brought forth a classification of phases of matter in terms of their ability to thermalize and to scramble quantum information. Thermalization of closed isolated quantum many-body systems is usually understood in terms of the ability of ergodic quantum many-body systems to act as an environment for themselves and therefore to bring to thermal equilibrium sufficiently local observables or subsystems  [3]. Scrambling on the other hand describes the possibility of hiding quantum information in non-local correlators and is related to operator spreading, quantum chaos and the exponential growth of certain Out-of-Time-Order Correlators [4–6]. Interest around these concepts has grown in a broad community across condensed matter, statistical physics and high-energy theory, motivated by the introduction of a minimal model for maximal chaos and fast scrambling, the Sachdev-Ye-Kitaev model ($SYK_4$) [7–14], describing Majorana fermions with random two-body all-to-all interactions. This models escapes the standard paradigm of quantum matter with quasiparticles, a statement that comes with profound implications for transport and thermalization times [15]. When brought out of equilibrium by a global excitation the $SYK_4$ model has been shown to thermalise under its own quantum dynamics with a rate that is essentially set by the temperature [16] and that can be controlled by adding to the $SYK_4$ model a relevant low-energy perturbation [17] or by considering the case of $q-$body interactions and the limit $q \to \infty$ in which thermalization is instantaneous [18].

Recently there has been large interest in the role of dissipation coming from an external environment on SYK physics. In the context of high-energy physics this has been motivated by questions about black-hole evaporation and wormholes [19–23] where the environment was modeled itself as a maximally chaotic $SYK_4$ system. In a condensed matter setting coupling to a non-interacting bath has been discussed to study transport and dynamics properties across Non-Fermi-Liquids to Fermi-Liquid transitions [24–27]. Finally, the role of Markovian dissipation on the chaotic properties of $SYK_4$ has been also discussed [28,29].

In this work we ask the question of whether a fast scrambler such as the $SYK_4$ model is also an efficient thermal bath for another quantum system coupled to it. In traditional approaches to open systems, from statistical mechanics to quantum optics, environments are usually treated as a macroscopic collection of non-interacting degrees of freedom with a continuum of excitations, as in the celebrated Caldeira-Legget problem [30]. Not much is therefore known about the role of interactions and chaoticity of the environment on the thermalization dynamics

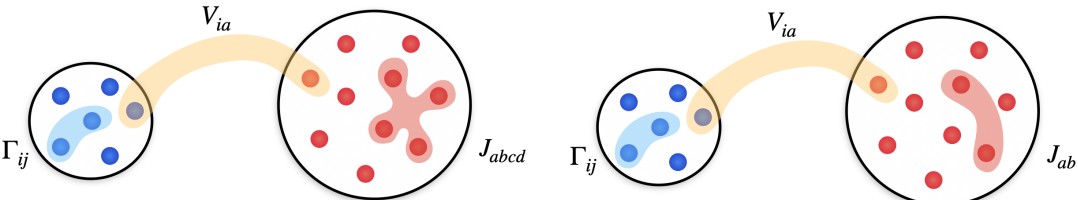

Figure 1: Sketch of the two settings we consider in this paper. *Left panel*: a system of $N$ Majorana fermions with random one-body couplings $\Gamma_{ij}$, described by the SYK$_2$ Hamiltonian, is coupled to an SYK$_4$ bath of $M \gg N$ Majorana fermions with random two-body couplings $J_{abcd}$. *Right panel*: same setting, but now the environment is described by $M \gg N$ non-interacting Majorana fermions with random couplings $J_{ab}$.

of the system. To address this question in a simple setting we consider the situation in which the system dynamics by its own is as simple as possible and all the interesting time-evolution comes from the coupling to the environment and from its properties. We choose therefore to model our system as a set of $N$ non-interacting randomly coupled Majorana fermions, referred to as SYK$_2$ model, which under their own dynamics would fail to thermalize due to lack of scattering. We couple this simple system to a much larger bath of $M \gg N$ Majorana modes with SYK$_4$ Hamiltonian and study the dissipative dynamics of the system induced by the environment. Using Keldysh techniques we study the thermalization dynamics of the system after a sudden switching of the system bath coupling. To highlight the role of bath interactions we compare this setting with a more conventional one where the environment is made by non-interacting degrees of freedom with a gapless spectrum. We show that the SYK$_4$ bath is more efficient in thermalising the system at weak coupling, due to its enhanced density of states at low frequency, while at strong system-bath couplings both type of environments give rise to a similar time scale for thermalisation.

The paper is structured as follows. In Sec. 2 we introduce the model(s) and the nonequilibrium setting under consideration and their large $N, M$ solution based on Keldysh techniques. In Sec. 3 we review the equilibrium properties of our models, focusing in particular on the spectral function of the SYK$_2$ system and how it is affected by the bath. In Sec. 4 we present our numerical results for the quench dynamics. We look at the thermalization of the spectral and distribution function of the system, the energy absorption rate and the dynamics of the effective temperature. In Sec. 5 we provide an analytical understanding of the regime of weak coupling to the bath using a Quantum Boltzmann Equation approach to the exact dynamics. We present our conclusions in Sec. 6. In the appendix we present additional analytical and numerical results in support of our analysis.

## 2  Model and its Large $N, M$ solution

We study a model of $N$ Majorana fermions $\chi_i$, describing our system, coupled to an environment made of $M \gg N$ Majorana fermions $\psi_a$ with total Hamiltonian (See Fig. 1)

$$H(t) = \frac{i}{2} \sum_{i,j=1}^{N} \Gamma_{i,j}\, \chi_i \chi_j + H_B[\psi] + \theta(t) H_{SB}[\chi, \psi] \tag{1}$$

Majorana fermions satisfy the anti-commutation relations $\{\chi_i, \chi_j\} = \delta_{ij}$ and $\{\psi_a, \psi_b\} = \delta_{ab}$. The small system is described by the SYK$_2$ model where $\Gamma_{ij}$ are random independent Gaussian variables with zero mean and variance $\overline{\Gamma_{ij}^2} = \frac{\Gamma^2}{N}$. We will compare the action of two different baths on the system: the $\psi$-fermions will follow either the SYK$_2$ model or the SYK$_4$ model

$$H_B^{(2)} = \frac{i}{2} \sum_{a,b=1}^{M} J_{ab}\, \psi_a \psi_b \tag{2}$$

$$H_B^{(4)} = -\frac{1}{4!} \sum_{a,b,c,d=1}^{M} J_{abcd}\, \psi_a \psi_b \psi_c \psi_d \tag{3}$$

We consider a linear coupling between the system and the bath

$$H_{SB} = i \sum_{i=1}^{N} \sum_{a=1}^{M} V_{ia}\, \chi_i \psi_a \tag{4}$$

$J_{ab}$, $J_{abcd}$ and $V_{ia}$ are random independent Gaussian variables with zero mean and variances $\overline{J_{ij}^2} = \frac{J^2}{M}$, $\overline{J_{ijkl}^2} = \frac{3!J^2}{M^3}$ and $\overline{V_{ia}^2} = \frac{V^2}{M}$ respectively. In Appendix C we will comment on the role of the form of the system-bath coupling in the dynamic of the system. This model has already been studied in [24, 27] but from a different perspective. The ratio $\epsilon = N/M$ was kept finite so the size of the two species of fermions was comparable. The authors observed a quantum phase transition between a fermi liquid and a non fermi liquid by varying the ratio $\epsilon$. The consequences on the dynamics of the two species of fermions was also discussed. In this paper we will focus on the limit $M \gg N$ to fashion an open quantum system setting.

We use Keldysh formalism [31] to obtain the non-equilibrium dynamics of the system after the quench. The partition function on the closed-time Keldysh contour $Z = \int \mathcal{D}[\chi, \psi] e^{iS[\chi,\psi]}$ is expressed in terms of the action $S[\chi, \psi]$

$$\begin{aligned}
S[\chi, \psi] = \int_{-\infty}^{+\infty} \sum_{\alpha=\pm} \alpha \Big\{ & \frac{i}{2} \sum_{i=1}^{N} \chi_i^\alpha(t) \partial_t \chi_i^\alpha(t) + \frac{i}{2} \sum_{a=1}^{M} \psi_a^\alpha(t) \partial_t \psi_a^\alpha(t) - \frac{i}{2} \sum_{i,j=1}^{N} \Gamma_{ij}\, \chi_i^\alpha \chi_i^\alpha \\
& + \frac{1}{4!} \sum_{a,b,c,d=1}^{M} J_{abcd}\, \psi_a^\alpha \psi_b^\alpha \psi_c^\alpha \psi_d^\alpha - i\, \theta(t) \sum_{i=1}^{N} \sum_{a=1}^{M} V_{ia}\, \chi_i^\alpha \psi_a^\alpha \Big\}
\end{aligned} \tag{5}$$

with $\alpha = \pm$ the Keldysh index denoting the upper and lower branches of the closed-time contour. We have written the action with the SYK$_4$ bath; it is straightforward to adapt to the

case of the $SYK_2$ bath. After averaging over the disorder we can rewrite the action in terms of the bilocal fields

$$G_S^{\alpha\beta}(t,t') = -\frac{i}{N}\sum_{i=1}^{N}\langle\chi_i^{\alpha}(t)\chi_i^{\beta}(t')\rangle = \begin{pmatrix} G_S^T(t,t') & G_S^<(t,t') \\ G_S^>(t,t') & G_S^{\tilde{T}}(t,t') \end{pmatrix}_{\alpha\beta} \tag{6}$$

$$G_B^{\alpha\beta}(t,t') = -\frac{i}{M}\sum_{a=1}^{M}\langle\psi_a^{\alpha}(t)\psi_a^{\beta}(t')\rangle \begin{pmatrix} G_B^T(t,t') & G_B^<(t,t') \\ G_B^>(t,t') & G_B^{\tilde{T}}(t,t') \end{pmatrix}_{\alpha\beta} \tag{7}$$

with the corresponding Lagrange multipliers

$$\Sigma_S^{\alpha\beta}(t_1,t_2) = \begin{pmatrix} \Sigma_S^T(t,t') & -\Sigma_S^<(t,t') \\ -\Sigma_S^>(t,t') & \Sigma_S^{\tilde{T}}(t,t') \end{pmatrix}_{\alpha\beta}, \qquad \Sigma_B^{\alpha\beta}(t_1,t_2) = \begin{pmatrix} \Sigma_B^T(t,t') & -\Sigma_B^<(t,t') \\ -\Sigma_B^>(t,t') & \Sigma_B^{\tilde{T}}(t,t') \end{pmatrix}_{\alpha\beta} \tag{8}$$

Integrating over the fermions $\chi$ and $\psi$ we are left with the effective action $S[G,\Sigma]$

$$\begin{aligned}
S[G,\Sigma] = &-i\frac{N}{2}\mathrm{Tr}\log\big[-i\hat{G}_{0,S}^{-1} + i\hat{\Sigma}_S\big] - i\frac{M}{2}\mathrm{Tr}\log\big[-i\hat{G}_{0,B}^{-1} + i\hat{\Sigma}_B\big] \\
&+ i\frac{N}{2}\int dt\, dt' \sum_{\alpha\beta}\alpha\beta\left(-\frac{\Gamma^2}{2}G_S^{\alpha\beta}(t,t')^2 + G_S^{\alpha\beta}(t,t')\Sigma_S^{\alpha\beta}(t,t')\right) \\
&+ i\frac{M}{2}\int dt\, dt' \sum_{\alpha\beta}\alpha\beta\left(i^{q_B}\frac{J^2}{q_B}G_B^{\alpha\beta}(t,t')^{q_B} + G_B^{\alpha\beta}(t,t')\Sigma_B^{\alpha\beta}(t,t')\right) \\
&- i\frac{N}{2}\int dt\, dt' \sum_{\alpha\beta}\alpha\beta\,\theta(t)\theta(t')V^2 G_S^{\alpha\beta}(t,t')\, G_B^{\alpha\beta}(t,t')
\end{aligned} \tag{9}$$

where $q_B = 2,4$ corresponds to the the $SYK_2$ and $SYK_4$ bath respectively. In the large $N$,$M$ limit the partition function is dominated by its saddle-point. Varying the action $S[G,\Sigma]$ with respect to $G$ and $\Sigma$ gives the Schwinger-Dyson equations

$$\left[\hat{G}_0^{-1} - \hat{\Sigma}_S\right]\circ\hat{G}_S = 1, \qquad \left[\hat{G}_0^{-1} - \hat{\Sigma}_B\right]\circ\hat{G}_B = 1 \tag{10}$$

and

$$\Sigma_S^{\alpha\beta}(t,t') = \alpha\beta\Gamma^2 G_S^{\alpha\beta}(t,t') + \alpha\beta V^2\,\theta(t)\theta(t')\, G_B^{\alpha\beta}(t,t') \tag{11}$$

$$\Sigma_B^{\alpha\beta}(t,t') = -i^{q_B}\alpha\beta J^2 G_B^{\alpha\beta}(t,t')^{q_B-1} + \epsilon\,\alpha\beta V^2\,\theta(t)\theta(t')\, G_S^{\alpha\beta}(t,t') \tag{12}$$

where $[\hat{G}_0^{-1}]^{\alpha\beta}(t,t') = i\alpha\delta_{\alpha\beta}\delta(t-t')\partial_t$ is the free Majorana Green's function.

It is worth commenting on the structure of the saddle-point Dyson equations and in particular of the self-energies, Eq. (11,12). First, we note that the self-energy of the system $\Sigma_S$ takes two contributions, one due to the random one-body coupling as in the isolated $SYK_2$ model and one coming from the coupling with the bath. The bath Green's function is also renormalized due to interaction processes, leading to a self-energy $\Sigma_B$. This takes both a contribution due to interactions within the bath (depending on the value of $q_B$) and a contribution due to the coupling between system and bath. This *feedback* of the system on the bath dynamics is

however suppressed by a small parameter $\epsilon = N/M \ll 1$, i.e. the ratio between the system and the bath size. In this work we focus on the limit $\epsilon \to 0$ in which the $\chi$ fermions of the system have no effect on the $\psi$ fermions in the bath and therefore we have

$$\Sigma_B^{\alpha\beta}(t,t') = -i^{q_B}\alpha\beta J^2 G_B^{\alpha\beta}(t,t')^{q_B-1} \tag{13}$$

Thus $G_B$ and $\Sigma_B$ will be fixed by the initial conditions during the whole process and we only focus on the dynamics of $G_S$. While we can consider that the $\psi$ fermions act as a bath for the $\chi$ fermions, we emphasize that the effective dynamics of the system is different depending on the nature of the environment. In fact a non-interacting $SYK_2$ bath can be formally integrated out exactly and gives rise to an effective action for the system that is quadratic, while this is in general not the case for an interacting $SYK_4$ bath. Finally, we note a similarity in the structure of the Dyson equations for our coupled SYK models with respect to the case of mixed SYK models, such as the $SYK_4+SYK_2$ model featuring both one-body and two-bodies random couplings. Also in that case in fact the self-energy gets contributions scaling as different powers of the Green's function, as in Eq. (11) where however it is crucial that the second term does not involve system degrees of freedom.

In order to solve numerically the Dyson equations it is convenient to manipulate and bring them to the form of Kadanoff-Baym equations. We define the retarded, advanced and Keldysh Green's function

$$G^R(t_1,t_2) \equiv \theta(t_1-t_2)\big(G^>(t_1,t_2) - G^<(t_1,t_2)\big) \tag{14}$$
$$G^A(t_1,t_2) \equiv -\theta(t_2-t_1)\big(G^>(t_1,t_2) - G^<(t_1,t_2)\big) \tag{15}$$
$$G^K(t_1,t_2) \equiv G^>(t_1,t_2) + G^<(t_1,t_2) \tag{16}$$

and likewise for the self-energies:

$$\Sigma^R(t_1,t_2) \equiv \theta(t_1-t_2)\big(\Sigma^>(t_1,t_2) - \Sigma^<(t_1,t_2)\big) \tag{17}$$
$$\Sigma^A(t_1,t_2) \equiv -\theta(t_2-t_1)\big(\Sigma^>(t_1,t_2) - \Sigma^<(t_1,t_2)\big) \tag{18}$$
$$\Sigma^K(t_1,t_2) = \Sigma^>(t_1,t_2) + \Sigma^<(t_1,t_2) \tag{19}$$

To get the Dyson equation in this basis, we can multiply equation (10) on the left and on the right by the unitary matrix

$$U = \frac{1}{\sqrt{2}}\begin{pmatrix} 1 & 1 \\ 1 & -1 \end{pmatrix} \tag{20}$$

which yields

$$\big(G_0^{-1} - \Sigma^R\big) \circ G^R = 1 \tag{21}$$
$$\big(G_0^{-1} - \Sigma^A\big) \circ G^A = 1 \tag{22}$$
$$\big(G_0^{-1} - \Sigma^R\big) \circ G^K = \Sigma^K \circ G^A \tag{23}$$

By linearity, the self-energies are simply

$$\Sigma_S^{R,A,K} = \Gamma^2 G_S^{R,A,K} + V^2 G_B^{R,A,K} \tag{24}$$

For Majorana fermions we have the relation between greater and lesser Green's function

$$G^>(t_1, t_2) = -G^<(t_2, t_1) \tag{25}$$

which allows to focus on the dynamics of $G_S^>(t_1, t_2)$ only. We rewrite the first Schwinger-Dyson equation in a form known as the Kadanoff-Baym equations:

$$i\partial_{t_1} G_S^>(t_1, t_2) = \int_{-\infty}^{+\infty} dt \left( \Sigma_S^R(t_1, t) G_S^>(t, t_2) + \Sigma_S^>(t_1, t) G_S^A(t, t_2) \right) \tag{26}$$

$$-i\partial_{t_2} G_S^>(t_1, t_2) = \int_{-\infty}^{+\infty} dt \left( G_S^R(t_1, t) \Sigma_S^>(t, t_2) + G_S^>(t_1, t) \Sigma_S^A(t, t_2) \right) \tag{27}$$

which is more convenient to work with. Using the property of the non-equilibrium Green's function $G^>(t_1, t_2) = -G^>(t_2, t_1)^*$ we only need to solve the first of the two equations.

Finally we introduce the energy of the system $E_S(t) = \langle H_S(t) \rangle$ where $\langle \cdots \rangle$ denotes the average over the disorder. It is straightforward to show that $E_S(t)$ is expressed in terms of $G_S^{>,<}$ as

$$E_S(t) = i \frac{\Gamma^2}{2} \int_{-\infty}^{t} dt' \left( G_S^>(t, t')^2 - G_S^<(t, t')^2 \right) \tag{28}$$

## 3 Equilibrium Properties

Before focusing on the dynamics of this model we briefly discuss its equilibrium properties. In equilibrium all two points correlators are only function of the time difference $t = t_1 - t_2$ so we can work with their Fourier transform

$$G(\omega) = \int_{-\infty}^{\infty} dt\, G(t) e^{i\omega t} \tag{29}$$

For future use, we remind that in the low energy limit $\omega, T_B \ll J$, the retarded Green's function of the bath is

$$G_B^R \simeq -\frac{i}{J} \quad (q_B = 2), \qquad G_B^R \simeq -\frac{i}{\sqrt{2\pi T_B}} \left( \frac{\pi}{J^2} \right)^{1/4} \frac{\Gamma\left( \frac{1}{4} - i\frac{\omega}{2\pi T_B} \right)}{\Gamma\left( \frac{3}{4} - i\frac{\omega}{2\pi T_B} \right)} \quad (q_B = 4) \tag{30}$$

In figure 2 (bottom panel) we plot the full spectral density of the bath $A_B(\omega) = -2\,\mathrm{Im}\,G_B^R(\omega)$ in the two cases $q_B = 2$ and $q_B = 4$.

In equilibrium the retarded Green's function of the system satisfies the Dyson equation

$$G_S^R(\omega)^{-1} = \omega - \Sigma_S^R(\omega), \qquad \Sigma_S^R(\omega) = \Gamma^2 G_S^R(\omega) + V^2 G_B^R(\omega) \tag{31}$$

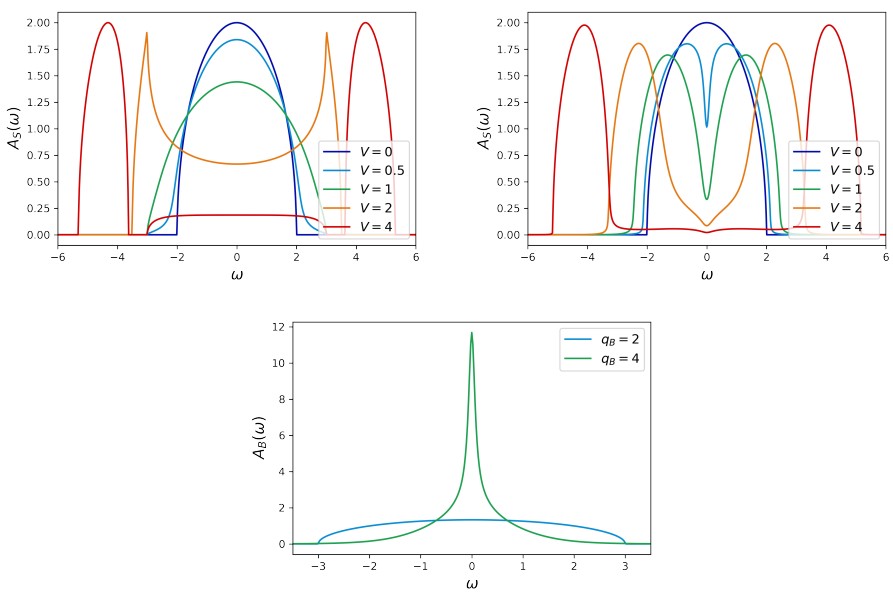

Figure 2: *Top panels:* spectral density of the system $A_S(\omega)$ for different couplings $V$ with $q_B = 2$ (*left*) and $q_B = 4$ (*right*). *Bottom panel*: spectral density of the bath $A_B(\omega)$ in the two cases $q_B = 2$ and $q_B = 4$. Parameters: $\Gamma = 1$, $J = 1.5$ and $\beta_B = 20$.

This is a quadratic equation on $G_S^R(\omega)$ and we give the exact solution to this equation in Appendix A. In figure 2 (top panels) we plot the spectral density of the system $A_S(\omega)$ when $q_B = 2$ and $q_B = 4$. We have assumed that the system is in thermal equilibrium at the temperature of the bath $T_B$. We notice that the low-energy behaviour is different in the two cases: when $q_B = 2$ the spectrum remains flat while for $q_B = 4$ there is a dip in the spectrum. The low-energy part of the spectrum can easily be obtained from the Dyson equation in the limit $\omega, T_B \ll \Gamma, J$. In this regime we can neglect the $\omega$ term and replace $G_B^R(\omega)$ by its conformal expression given above. If the bath is $\mathrm{SYK}_2$ we get

$$G_S^R(\omega) = -\frac{i}{\tilde{\Gamma}}, \qquad \tilde{\Gamma} = \frac{\Gamma}{\sqrt{1 + \left(\frac{V^2}{2\Gamma J}\right)^2} - \frac{V^2}{2\Gamma J}}, \qquad (q_B = 2) \qquad (32)$$

Thus, the system still behaves like an $\mathrm{SYK}_2$ model at low energy but with a coupling constant renormalized by the bath. If now the bath is $\mathrm{SYK}_4$, the low energy behaviour is found by neglecting also the term $\Gamma^2 G_S^R(\omega)$. The Dyson equation reduces to $G_S^R(\omega) G_B^R(\omega) = -1/V^2$ giving

$$G_S^R(\omega) = -\frac{i}{V^2}\left(\frac{J^2}{\pi}\right)^{\frac{1}{4}} \sqrt{2\pi T_B} \frac{\Gamma\left(\frac{3}{4} - i\frac{\omega}{2\pi T_B}\right)}{\Gamma\left(\frac{1}{4} - i\frac{\omega}{2\pi T_B}\right)}, \qquad (q_B = 4) \qquad (33)$$

We see that the dip in the spectral density $A_S(\omega)$ is due to the sharp peak at low frequency in the spectral density of the bath $A_B(\omega)$. At finite temperature, this solution is valid provided that $\Gamma^2 G_S^R(\omega) \ll V^2 G_B^R(\omega)$ i.e $V^4 \gg \Gamma^2 J T_B$. If this is not the case, we can treat $V^2 G_B^R(\omega)$ in

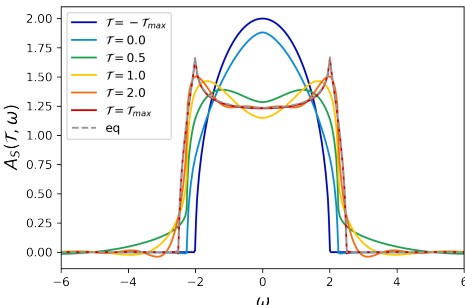 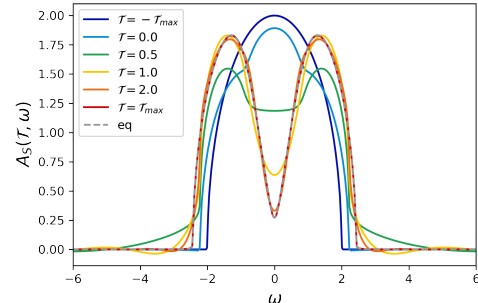

Figure 3: Evolution of the spectral density of the system after the quench for $q_B = 2$ (*left panel*) and $q_B = 4$ (*right panel*). The dotted grey line is the equilibrium Green's function at the temperature of the bath. Parameters: $V = 1$, $\beta_B = 20$.

the Dyson equation as a perturbation around the SYK$_2$ conformal solution $G_S^{R,0}(\omega) = -i/\Gamma$. At first order in $V^2/\Gamma^2$, we get

$$G_S^R(\omega) = -\frac{i}{\Gamma} - \frac{V^2}{2\Gamma^2}G_B^R(\omega) + O\left(\frac{V^4}{\Gamma^4}\right) \tag{34}$$

We also find in the regime a dip at low frequency in the system spectral density $A_S(\omega)$.

As we increase $V$, we see a gap opening in the spectral density $A_S(\omega)$, with two symmetric peaks which concentrate the spectral weight. This happens with both the SYK$_2$ and the SYK$_4$ baths, and the spectral densities in the two cases start to look more alike as we increase $V$. We show in Appendix A that for large $V$ the peak is located around $\omega_0 = V$ and has a width $\Delta = 2\Gamma + O(\Gamma/V)$ for $V \gg \Gamma$.

# 4 Quench dynamics after a sudden switching of the system-bath coupling: numerical solution

In this section we present our results for the quench dynamics of the model. Initially the system and the bath are decoupled and are prepared in a thermal state at temperature $T_S$ and $T_B$ respectively with $T_B < T_S$. At time $t = 0$, we switch on the coupling between the two and we look at the effect of the interaction on the dynamics of the small system. Like we said before, the bath is considered to be at thermal equilibrium during the whole process, so $G_B^>$ is computed from the equilibrium solution $G_B^{>,<}(t_1, t_2) = G_{B,eq}^{>,<}(t_1 - t_2)$. If not stated otherwise the inverse temperature of the bath is fixed at $\beta_B = 20$. To get the out-of-equilibrium dynamics of the small system we solve the Kadanoff-Baym equations numerically. We use a $t_1 - t_2$ grid of size $4001 \times 4001$ with time step $dt = 0.05$. The length in each direction is $2t_{max}$. Initially at times $t_1, t_2 < 0$ the system is prepared in a thermal state of the SYK$_2$ model at inverse temperature $\beta_S = 10$. The Kadanoff-Baym equations are solved on the grid using a predictor-corrector scheme [17]. We chose for the coupling strenghts of the system and the bath $\Gamma = J = 1$. Thermal equilibrium Green's function are computed by solving the

Schwinger-Dyson equation self-consistently as in [32].

## 4.1    Thermalisation Dynamics: Spectral and Distribution Functions

It is common when working with real time two-point correlators to change coordinates $(t_1, t_2)$ to central and relative times $(\mathcal{T}, t) = (\frac{t_1 + t_2}{2}, t_1 - t_2)$. Then we can define the so-called Wigner transform which is the Fourier transform with respect to the time difference $t$:

$$G(\mathcal{T}, \omega) = \int dt\, e^{i\omega t} G\Big(t_1 = \mathcal{T} + \frac{t}{2}, t_2 = \mathcal{T} - \frac{t}{2}\Big) \tag{35}$$

If after the quench the system reaches equilibrium, two-point functions should depend only on the time difference $t$ and no longer on the central time $\mathcal{T}$. In our numerical simulations $\mathcal{T}$ varies between $-\mathcal{T}_{max}$ and $\mathcal{T}_{max}$ with $\mathcal{T}_{max} = t_{max}/2$.

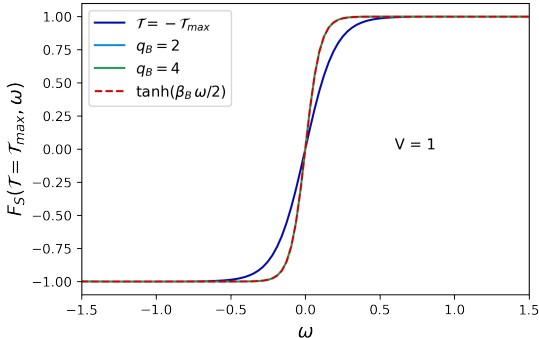

Figure 4: Distribution function of the system long after the quench. We show the result with the two baths $q_B = 2$ (light blue line) and $q_B = 4$ (green line). The dotted red line is the equilibrium distribution function at the temperature of the bath $T_B$.

We will follow the evolution of the spectral density $A(\mathcal{T}, \omega)$ and the distribution function $F(\mathcal{T}, \omega)$

$$A(\mathcal{T}, \omega) = -2\mathrm{Im}G^R(\mathcal{T}, \omega), \qquad iG^K(\mathcal{T}, \omega) = F(\mathcal{T}, \omega)A(\mathcal{T}, \omega) \tag{36}$$

If the system reaches thermal equilibrium, by the fluctuation dissipation theorem, the distribution function should become $F(\omega) = \tanh\left(\frac{\beta\omega}{2}\right)$ where $\beta$ is the inverse temperature.

Figure 3 shows the evolution of the spectral density $A_S(\mathcal{T}, \omega)$ of the system after the quench. We can see how the gap opens in the dynamics. Besides, we see that long after the quench the spectral density reaches the thermal spectral density computed in the previous section at the temperature of the bath (dotted gray line). In figure 4 we show that long after the quench the distribution function coincides with $F_{eq}(\omega) = \tanh\left(\frac{\beta_B\omega}{2}\right)$ which is another indication that the system relaxes to a thermal state at the temperature of the bath $T_B$ long after the quench.

We want to stress here that the fact that the SYK$_2$ system reaches thermal equilibrium after the quench is not enterily trivial. Indeed, it was shown in Ref. [16] that in an isolated

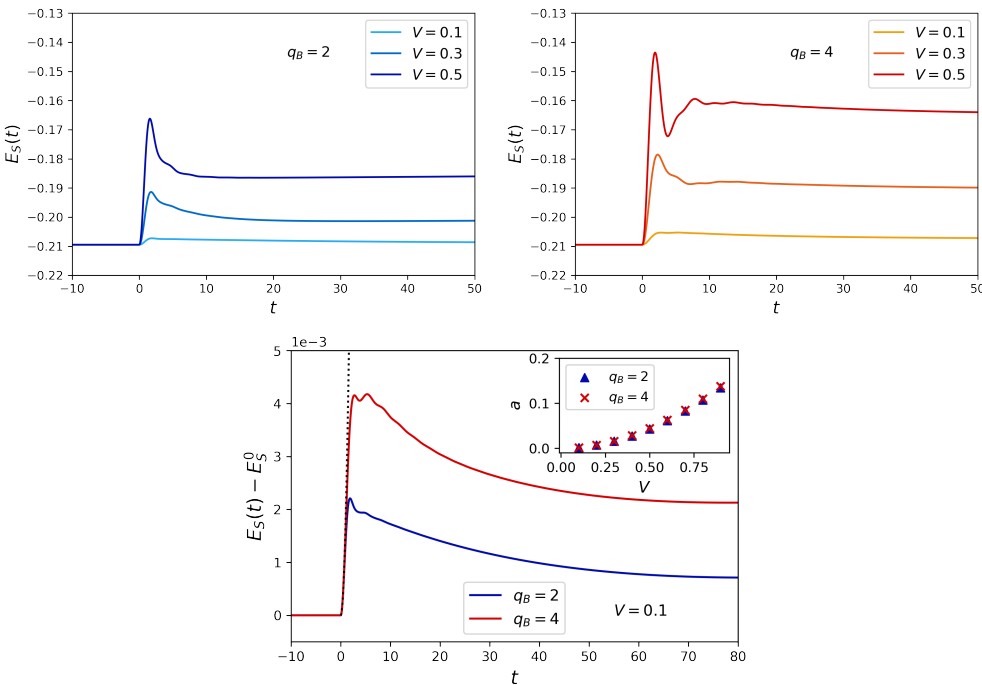

Figure 5: Energy of the system after the quench. *Top left panel*: $q_B = 2$. *Top right panel*: $q_B = 4$. *Bottom panel*: Comparison of the energy of the system $E_S(t)$ in the two cases $q_B = 2$ and $q_B = 4$ for $V = 0.1$. The dotted grey line is a quadratic fit $a_{q_B} t^2$. In the inset we plot $a_{q_B}$ as a function of $V$. We see that the coefficient is the same with the two baths and depends quadratically on $V$.

setting, if the final theory after a quantum quench is $SYK_2$, the system reaches equilibrium instantaneously and the final state is not thermal. The fact that the equilibration is sudden can be seen from the Kadanoff-Baym equations (26) (27) by replacing $\Sigma = J^2 G$ and taking the difference between the two equations, which gives $\partial_{\mathcal{T}} G^{>}(\mathcal{T}, t) = 0$. Furthermore, in Ref. [27] the authors showed that when $N$ $SYK_2$ fermions are linearly coupled to $M$ $SYK_2$ fermions after a quench with a finite ratio $N/M$, the $SYK_2$ fermions fail to thermalize. Here we have shown that in the limit $N/M \to 0$ i.e when coupled to an (isolated) bath, the $SYK_2$ fermions do reach thermal equilibrium at the temperature of the bath $T_B$ after a quench. This result is valid both with the $SYK_2$ and the $SYK_4$ bath.

## 4.2   Thermalisation Dynamics: Energy and Effective Temperature

In the previous section we have shown that the small $SYK_2$ system reaches thermal equilibrium when it is coupled to a thermal bath. Now we would like to characterise the approach to equilibrium and compare the evolution with the $q_B = 2$ and $q_B = 4$ baths.

In figure 5 we plot the evolution of the energy of the system $E_S(t)$ after the quench. Before the quench, the system is hotter than the bath so we expect that energy flows from the system to the bath in order for the system to cool down and reach the temperature of the bath. As already pointed out in Ref. [20], right after the quench the system first absorbs energy form

the bath: the coupling creates excitations in the system which will later decay into the bath. Comparing the energy long before and after the quench in figures 5 (top panels), we see that overall the system has gained energy through the quench process. Naturally the energy gain increases with $V$. Besides, the system absorbs more energy when it is coupled to the $SYK_4$ bath compared to the $SYK_2$ bath. In Ref. [20] the authors showed by doing a perturbative calculation in $V$ that at short time energy increases quadratically $E_S(t) = E_S^0 + a_{q_B}t^2$. Our numerical simulation also shows this initial quadratic increase (dotted grey line on figure 5 bottom panel). We also find that $a_{q_B}$ is proportional to $V^2$ and is very similar with the two baths (see inset).

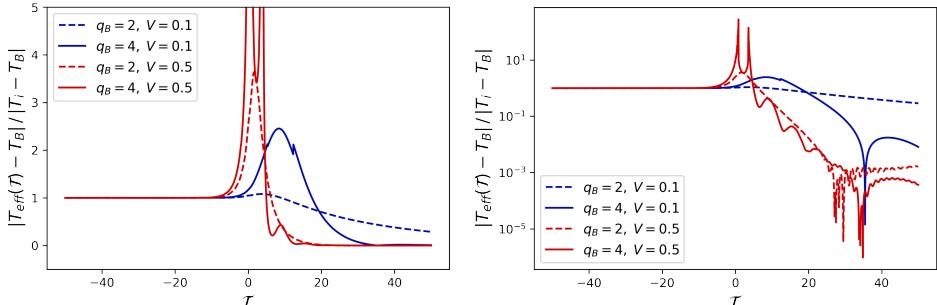

Figure 6: Effective temperature of the system $T_{eff}(\mathcal{T})$ in the two cases $q_B = 2$ and $q_B = 4$ with $V = 0.1$ and $V = 0.5$. The *right panel* is the same plot as the *left panel* but in log scale to underline the exponential scaling.

A convenient way to follow the dynamics of the system is to introduce an effective temperature $T_{eff}(\mathcal{T}) = 1/\beta_{eff}(\mathcal{T})$. We can proceed by generalising the fluctuation dissipation relation to non equilibrium settings and parametrizing the distribution function $F$ as

$$F(\mathcal{T}, \omega) \equiv \tanh\left(\frac{\beta_{eff}(\mathcal{T})\omega}{2}\right) \tag{37}$$

This allows to follow the approach of the system towards thermal equilibrium. In practice, we extract $\beta_{eff}(\mathcal{T})$ by fitting the low frequency part of $F(\mathcal{T}, \omega)$ to $\tanh\left(\frac{\beta_{eff}(\mathcal{T})\omega}{2}\right)$. The result is plotted in figure 6 in normal scale (left panel) and in log scale (right panel) for two values of the coupling $V$. We see that after the quench the system first heats up before cooling down. This is consistent with the dynamics of the energy discussed above as the system first absorbs energy before relaxing. Besides, the heating up after the quench increases with $V$ and is significantly more important with the $SYK_4$ bath than with the $SYK_2$ bath. After this initial heating part, we see on figure 6 that the decay of the effective temperature $T_{eff}(\mathcal{T})$ towards the temperature of the bath $T_B$ is exponential

$$T_{eff}(\mathcal{T}) = T_B + (T_S - T_B)e^{-\lambda\mathcal{T}} \tag{38}$$

For $V = 0.1$, we see that thermalization is much faster with the $SYK_4$ bath whereas for $V = 0.5$, apart from the oscillations, the decay looks very similar with the two baths. This resembles to what happens in equilibrium: as $V$ increases the spectral density of the system $A_S(\omega)$ is analogous when $q_B = 2$ or $q_B = 4$ with the spectral weight concentrated in two peaks

as $w = \pm V$. In the next section we will focus on the small $V$ regime where the differences in the dynamics with the two baths are more pronounced. Using the Quantum Boltzmann Equation (QBE) we will get an expression of the exponential decay rate $\lambda$ in this regime and show that thermalisation is indeed faster with the $SYK_4$ bath compared to the non-interacting $SYK_2$ bath.

# 5   Quantum Boltzmann Equation

The Boltzmann equation is an approximate equation describing the evolution of the distribution function $F_S(\mathcal{T}, \omega)$. The starting point is the (exact) Dyson equation on the Keldysh Green's function $G_S^K$ (23). We parametrize the Keldysh Green's function $G_S^K$ with the distribution function $F_S$ according to $G_S^K = G_S^R \circ F_S - F_S \circ G_S^A$ (likewise for the bath). One can recast the Dyson equation into [31]

$$\left(\partial_{t_1} + \partial_{t_2}\right)F_S = i\Sigma_S^K - i\left(\Sigma_S^R \circ F_S - F_S \circ \Sigma_S^A\right) \tag{39}$$

The next step is to take the Wigner transform of this kinetic equation. We need to take care of the time convolution $\circ$. If $g$ and $h$ are two-point functions, the Wigner transform of their convolution $f = g \circ h$ is [31]

$$f(\mathcal{T}, \omega) = g(\mathcal{T}, \omega)e^{-\frac{i}{2}(\overleftarrow{\partial_\mathcal{T}}\overrightarrow{\partial_\omega} - \overleftarrow{\partial_\omega}\overrightarrow{\partial_\mathcal{T}})}h(\mathcal{T}, \omega) \tag{40}$$

$$= g(\mathcal{T}, \omega)h(\mathcal{T}, \omega) - \frac{i}{2}\left[\partial_\mathcal{T}g\partial_\omega h - \partial_\omega g\partial_\mathcal{T}h\right] + \cdots \tag{41}$$

We made a gradient expansion in the second line. To see when this approximation is valid we can introduce $\delta t$ and $\delta\omega$ the typical scales on which the Green's functions $G$ vary

$$\frac{G(\mathcal{T}, \omega)}{\partial_\mathcal{T}G(\mathcal{T}, \omega)} \sim \delta t, \qquad \frac{G(\mathcal{T}, \omega)}{\partial_\omega G(\mathcal{T}, \omega)} \sim \delta\omega \tag{42}$$

Then the gradient expansion requires that

$$\delta t\, \delta\omega \gg 1 \tag{43}$$

We will comment below on this approximation and give further details in Appendix B. Keeping only the zeroth order term in the gradient expansion we get

$$\partial_\mathcal{T}F_S(\mathcal{T}, \omega) = i\Sigma_S^K(\mathcal{T}, \omega) - iF_S(\mathcal{T}, \omega)\left(\Sigma_S^R(\mathcal{T}, \omega) - \Sigma^R(\mathcal{T}, \omega)\right) \tag{44}$$

Replacing the self-energies by their expressions (24) we have after simplification

$$\partial_\mathcal{T}F_S + \left[F_S(\mathcal{T}, \omega) - \tanh\left(\frac{\beta_B\omega}{2}\right)\right]V^2A_B(\omega) = 0 \tag{45}$$

where we assumed that the bath satisfies FDT

$$iG_B^K(\omega) = \tanh\left(\frac{\beta_f\omega}{2}\right)A_B(\omega) \tag{46}$$

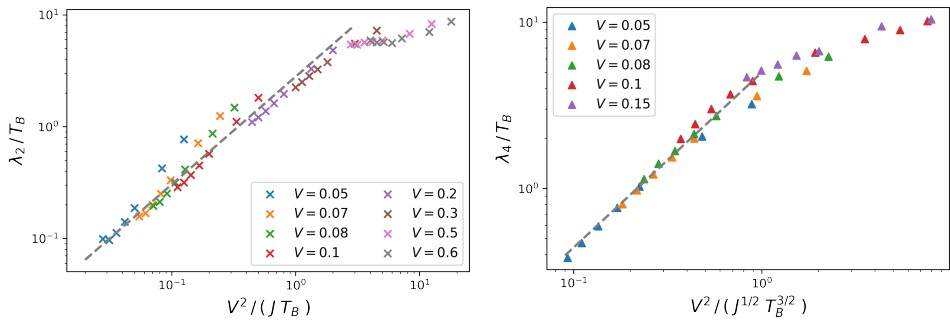

Figure 7: Temperature decay rate $\lambda$ for $q_B = 2$ (*left panel*) $q_B = 4$ (*right panel*). In these plots we vary both the coupling $V$ and the temperature of the bath $0.02 \leq T_B \leq 0.09$. The dotted grey line is a fit $y = \alpha \log(x) + \beta$. We find $\alpha = 0.96$ and $\alpha = 1.05$ for $q_B = 2$ and $q_B = 4$ respectively. Thus in both cases $\lambda_i / T_B$ scales linearly with $\epsilon_i$.

The solution to this equation is

$$F_S(\mathcal{T}, \omega) = \tanh\left(\frac{\beta_f \omega}{2}\right) + \left[F_S(\mathcal{T}_0, \omega) - \tanh\left(\frac{\beta_f \omega}{2}\right)\right] e^{-\lambda(\omega)(\mathcal{T} - \mathcal{T}_0)} \tag{47}$$

with

$$\lambda(\omega) = V^2 A_B(\omega) \tag{48}$$

Therefore, the QBE predicts that the distribution function approaches the thermal distribution $\tanh\left(\frac{\beta_f \omega}{2}\right)$ exponentially, meaning that the system relaxes to thermal equilibrium at the temperature of the bath. To get the dynamics of the effective temperature, we can parametrize $F_S(\mathcal{T}, \omega) = \tanh\left(\frac{\beta(\mathcal{T})\omega}{2}\right)$ and expand to first order in $\omega$ for $\omega \ll T_f$. We find

$$\beta(\mathcal{T}) = \beta_f + (\beta_0 - \beta_f) e^{-\lambda(0)(\mathcal{T} - \mathcal{T}_0)} \tag{49}$$

Thus, the effective temperature decays exponentially with a rate

$$\lambda_2 = 2\frac{V^2}{J}, \qquad \lambda_4 = \frac{1}{(4\pi)^{1/4}} \frac{\Gamma(1/4)}{\Gamma(3/4)} \frac{V^2}{\sqrt{JT_B}} \tag{50}$$

with the SYK$_2$ and SYK$_4$ baths respectively. As we are in the conformal limit $T_B \ll J$, we conclude that the dynamics is much faster with the SYK$_4$ bath

$$\lambda_4 \gg \lambda_2 \tag{51}$$

Let's come back to the domain of validity of the Boltzmann equation. We will show in Appendix B that the gradient expansion is accurate when

$$\begin{aligned} q_B = 2: \quad &\frac{V^2}{J^2} \ll 1 \\ q_B = 4: \quad &\frac{V^2}{J^{1/2} T_B^{3/2}} \ll 1 \end{aligned} \tag{52}$$

We can understand this criterion qualitatively. As the evolution is exponential, the typical time scale over which the dynamics takes place is $\delta t \simeq 1/\lambda$. Thus the gradient expansion works when $\lambda \ll \delta\omega$. Furthemore, $\delta\omega$ must be the typical energy scale on which the bath Green's function vary, because it is the bath which is responsible for the dynamics (remember that an isolated $SYK_2$ system equilibriates instantaneously). For $SYK_2$, we get $\delta\omega \sim J$ while for $SYK_4$ it is $\delta\omega \sim T_B$. Using $\lambda = V^2 A_B$ we get the criterion mentionned above.

We can test the prediction of the QBE approach against our numerical simulation. From the evolution of the effective temperature we can fit an exponential function and extract a decay rate. We plot in figure 7

$$\frac{\lambda_i}{T_B} = f\left(\frac{\epsilon_i}{T_B}\right), \qquad \epsilon_2 = \frac{V^2}{J}, \qquad \epsilon_4 = \frac{V^2}{\sqrt{JT_B}} \tag{53}$$

When $\epsilon_i \ll T_B$, according to the QBE analysis the scaling function $f$ should be linear. We see on figure 7 that this is indeed what we observe.

## 6 Discussion and Conclusion

In this work we have addressed the question of whether and under which conditions a fast scrambler such as the SYK model can act efficiently as thermal environment for another quantum system. Specifically we have studied the nonequilibrium dynamics of a system of non-interacting random Majorana fermions, the so called $SYK_2$ model, after a sudden switching of the coupling to an $SYK_4$ bath and compared the results to a more conventional bath made by non-interacting degrees of freedom with continuous spectrum. The choice of the $SYK_2$ as our *system* is due to both its simplicity and to the fact that if isolated this model is known to fail to thermalize due to its non-interacting nature.

Solving the Kadanoff-Baym equations numerically we have shown that the $SYK_2$ system reaches thermal equilibrium with the environment independently from the nature of the latter, as demonstrated by looking at its spectral and distribution function. Furthermore we have shown that the rate of energy absorption at short times, given by a Fermi-Golden-Rule type of result is essentially independent on the details of the bath. However the long time dynamics and the time-scales for thermalization depend strongly on the environment. In particular we have shown that for a weak coupling to the environment the $SYK_4$ bath is able to thermalize the system much faster, while upon increasing the coupling strength to the bath thermalization happens on similar time scales. In order to gain insights on the regime of weak-coupling we have analyzed the exact Dyson equations using a Quantum Boltzmann Equation type of expansion. This has allowed us to demonstrate that the approach to thermal equilibrium for the dynamics of the effective temperature is exponential and to estimate the scaling of the thermalization rate for the case of an $SYK_2$ and $SYK_4$ bath. Interestingly we have shown that the fast scrambling nature of the bath leaves an imprint on the system dynamics in terms of a thermalization rate $\lambda \sim \sqrt{T_B}$ in the range of temperatures $V^{4/3}/J^{1/3} \ll T_B \ll J$, as opposed to the constant rate expected for a good Markovian bath such as the $SYK_2$.

This work suggests a number of extensions that we plan to address in the near future. From one side it would be interesting to explore in more details the effect of the system feedback over the bath dynamics, due to a finite size of the environment. Another intriguing question

concerns whether the rich low-energy properties of the $SYK_4$ bath show up and affect other physical properties of the system other than its thermalization rate.

## Acknowledgements

**Author contributions**   AL developed and run the code for the dynamics of the $SYK_2$ system coupled to a bath.  All the authors conceived the projects and contributed to the writing of the manuscript.

**Funding information**   MS and SJT acknowledge support from the ANR grant "NonEQuMat" (ANR-19-CE47-0001) and computational resources on the Collège de France IPH cluster.

## A   Appendix : Retarded Green's function at equilibrium

### A.1   Exact solution

In this appendix we derive the exact expression of the system spectral density $A_S(\omega)$ in terms of the bath spectral density $A_B(\omega)$. As $\Sigma_S^{>,<}$ is a linear function of the system and bath Green's functions

$$\Sigma_S^{>,<}(t) = \Gamma^2 G_S^{>,<}(t) + V^2 G_B^{>,<}(t) \tag{54}$$

the retarded self-energy $\Sigma^R(t) = \theta(t)\big(\Sigma^>(t) - \Sigma^<(t)\big)$ takes the same form

$$\Sigma_S^R(t) = \Gamma^2 G_S^R(t) + V^2 G_B^R(t) \tag{55}$$

Taking the Fourier transform and plugging into the Dyson equation $G_S^R(\omega)^{-1} = \omega - \Sigma_S^R(\omega)$ we get

$$\Gamma^2 G_S^R(\omega)^2 - \big(\omega - V^2 G_B^R(\omega)\big)G_S^R(\omega) + 1 = 0 \tag{56}$$

This is a quadratic equation on $G_S^R(\omega)$ which is solved by computing its discriminant and its (complex) square root. In particular, the system spectral density $A_S(\omega) = -2\mathrm{Im}\,G^R(\omega)$ is given by

$$A_S(\omega) = \frac{1}{\Gamma^2}\sqrt{\frac{1}{2}\Big(\sqrt{u(\omega)^2 + v(\omega)^2} - u(\omega)\Big)} - \frac{V^2}{2\Gamma^2}A_B(\omega) \tag{57}$$

where

$$u(\omega) = \omega^2 - 4\Gamma^2 - 2\omega V^2 \mathrm{Re}\,G_B^R(\omega) + V^4\Big(\big(\mathrm{Re}\,G_B^R(\omega)\big)^2 - \big(\mathrm{Im}\,G_B^R(\omega)\big)^2\Big) \tag{58}$$

$$v(\omega) = -2V^2 \mathrm{Im}\,G_B^R(\omega)\Big(\omega - V^2 \mathrm{Re}\,G_B^R(\omega)\Big) \tag{59}$$

## A.2    Location of the peak for large $V$

In this section we try to locate the peak of the spectral density for $V \gg \Gamma, J$. If we look at frequencies $\omega$ close to the peak, we can assume that $G_B^R(\omega)$ is well-approximated by its large frequency behaviour $G_B^R(\omega) \sim 1/\omega$. Using the notations of the previous section we have

$$A_S(\omega) = \frac{1}{\Gamma^2} \sqrt{\frac{1}{2} \Big( |u(\omega)| - u(\omega) \Big)}, \qquad u(\omega) = 4\Gamma^2 - \omega^2 + 2V^2 - \frac{V^4}{\omega^2} \tag{60}$$

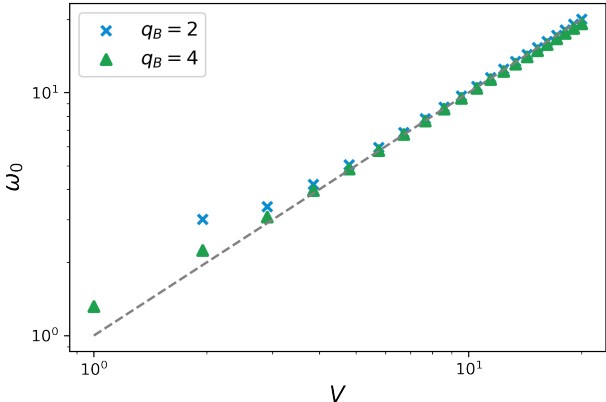

Figure 8: Position $\omega_0$ of the peak extracted numerically from the spectral density $A_S(\omega)$ as a function of $V$.

Thus in the region, $A_S(\omega)$ is non zero only if $u(\omega) < 0$. This is the case if $\omega_- < \omega < \omega_+$ where

$$\begin{aligned} \omega_\pm &= V \sqrt{1 + 2\frac{\Gamma^2}{V^2} \pm 2\frac{\Gamma}{V} \sqrt{1 + \frac{\Gamma^2}{V^2}}} \\ &\simeq V\Big(1 \pm \frac{\Gamma}{V}\Big) + O\Big(\frac{\Gamma^2}{V^2}\Big) \end{aligned} \tag{61}$$

In the last line we assumed $V \gg \Gamma$. In this interval $A_S(\omega)$ is given by

$$A_S(\omega) = \frac{1}{\Gamma^2} \sqrt{4\Gamma^2 - \omega^2 + 2V^2 - \frac{V^4}{\omega^2}} \tag{62}$$

One find that the (positive frequency) maximum of this function is located at $\omega_0 = V$, and the width of the peak is $\Delta = \omega_+ - \omega_- = 2\Gamma + O(\Gamma/V)$.

# B    Appendix : Validity of the gradient expansion

In this appendix we discuss the validity of the QBE approach. We can compare the magnitude of the zeroth and first order terms in the gradient expansion to find the domain where the

gradient expansion can be applied. At first order, equation (45) is replaced by

$$\left(1 - V^2 \frac{d\mathrm{Re}G_B^R}{d\omega}\right) \frac{\partial F_S}{\partial \mathcal{T}} + \left[F_S(\mathcal{T}, \omega) - \tanh\left(\frac{\beta_B \omega}{2}\right)\right] V^2 A_B(\omega) = 0 \tag{63}$$

The solution to this equation is still (47) with $\lambda(\omega)$ replaced by

$$\lambda(\omega) = \frac{V^2 A_B(\omega)}{1 - V^2 \frac{d\mathrm{Re}G_B^R}{d\omega}} \tag{64}$$

Thus the gradient expansion is valid provided that

$$V^2 \frac{d\mathrm{Re}G_B^R}{d\omega} \ll 1 \tag{65}$$

## B.1  $q_B = 2$

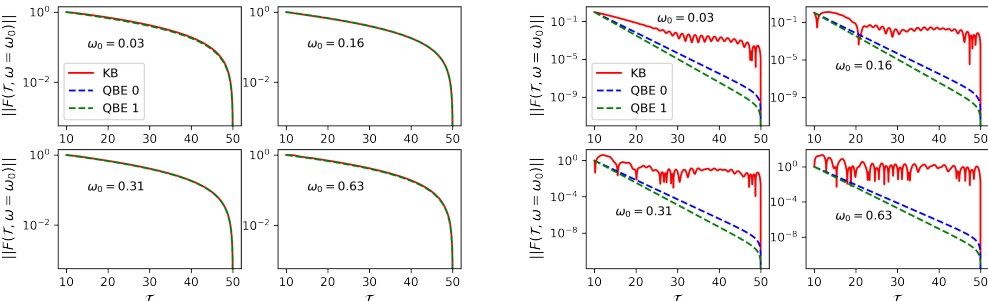

Figure 9: (Normalized) distribution function $||F(\mathcal{T}, \omega_0)||$ as a function of central time $\mathcal{T}$ for different values of $\omega_0$ after a quench with the $q_B = 2$ bath. We compare the solution of the KB equation (solid red line) with the one of the QBE at zero order (dotted blue line) and first order (dotted green line) in the gradient expansion. The *left panel* is for $V = 0.1$, for which the agreement between the two methods is good, and the *right panel* is for $V = 0.5$ for which the QBE fails to capture the dynamics. In both cases temperature of the bath is set to $T_B = 0.05$ and $\mathcal{T}_0 = 10$.

The retarded Green's of the SYK$_2$ bath is

$$G_B^R(\omega) = \frac{1}{2J^2}\left[\omega - i\sqrt{4J^2 - \omega^2}\right], \qquad |\omega| < 2J \tag{66}$$

and so

$$\frac{d\mathrm{Re}G_B^R}{d\omega} = \frac{1}{2J^2} \tag{67}$$

Therefore the gradient expansion is valid if

$$V^2 \ll 2J^2 \qquad\qquad q_B = 2 \tag{68}$$

In figure 9 we plot the (normalized) distribution function

$$||F(\mathcal{T},\omega)|| = \left| \frac{F(\mathcal{T},\omega) - F(\infty,\omega)}{F(\mathcal{T}_0,\omega) - F(\infty,\omega)} \right|, \qquad \mathcal{T} \geq \mathcal{T}_0 \tag{69}$$

at fixed $\omega = \omega_0$ for different coupling $V$. We see that the agreement is better with $V = 0.1$ than with $V = 0.5$ where it fails to reproduce the dynamics. Besides, we don't see any difference by varying $\omega_0$ for a fixed value of $V$.

## B.2  $q_B = 4$

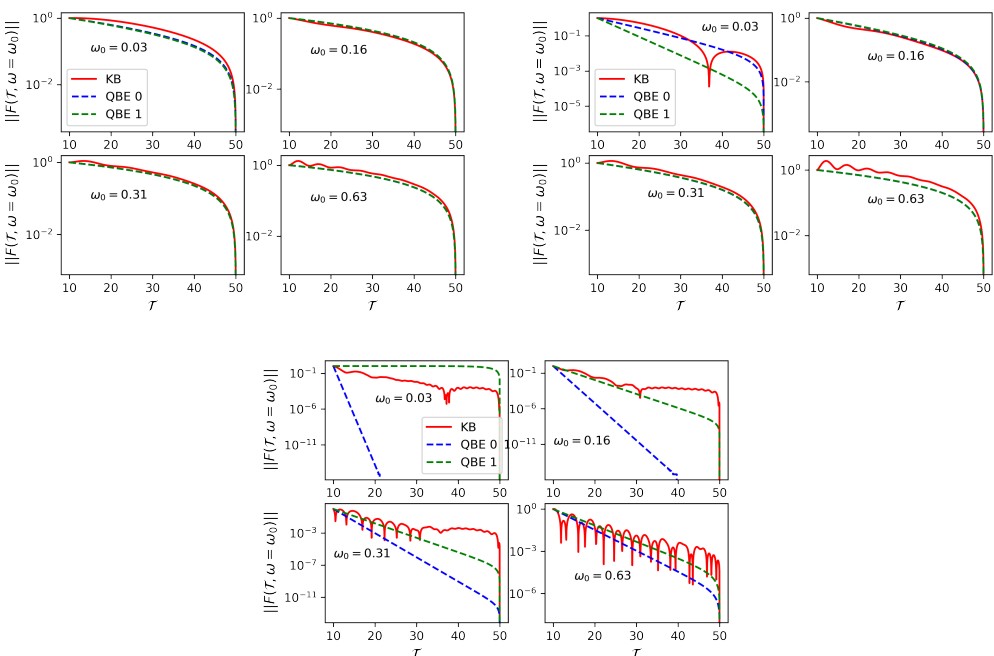

Figure 10: (Normalized) distribution function $||F(\mathcal{T},\omega_0)||$ as a function of central time $\mathcal{T}$ for different values of $\omega_0$ after a quench with the $q_B = 4$ bath. We compare the solution of the KB equation (solid red line) with the one of the QBE at zero order (dotted blue line) and first order (dotted green line) in the gradient expansion. The *top left*, *top right* and *bottom panels* are respectively for $V = 0.05$, $V = 0.1$ and $V = 0.5$. Temperature of the bath is set to $T_B = 0.05$ and $\mathcal{T}_0 = 10$. As discussed above, the agreement gets better as $\omega_0$ increases. For what concerns the low frequency part, the QBE is accurate only for very small $V$, like $V = 0.05$ as expected.

In the conformal limit $\omega, T \ll J$ the real part of the retarded Green's function of the $q_B = 4$ bath is

$$\mathrm{Re}G_B^R(\omega) = \left( \frac{\pi}{J^2} \right)^{1/4} \frac{1}{\sqrt{2\pi T}} \mathrm{Im}\left( \frac{\Gamma\left(\frac{1}{4} - i\frac{\omega}{2\pi T}\right)}{\left(\frac{3}{4} - i\frac{\omega}{2\pi T}\right)} \right) \tag{70}$$

We can distinguish two cases. If $\omega \ll T \ll J$, then $\mathrm{Re}G_B^R(\omega)$ reduces to a linear function of frequency

$$\mathrm{Re}G_B^R(\omega) \propto \frac{\omega}{J^{1/2}T^{3/2}} \tag{71}$$

where we omitted the numerical prefactors. Then the QBE is valid at low frequency provided that

$$V^2 \ll J^{1/2}T^{3/2} \qquad \omega \ll T \ll J \qquad (q_B = 4) \tag{72}$$

Now if $T \ll \omega \ll J$ we can take the zero-temperature limit of $\mathrm{Re}G_B^R(\omega)$ and we get

$$\mathrm{Re}G_B^R(\omega) \propto \frac{1}{\sqrt{J\omega}} \tag{73}$$

The criterion in this intermediate range of frequencies is

$$V^2 \ll J^{1/2}\omega^{3/2} \qquad T \ll \omega \ll J \qquad (q_B = 4) \tag{74}$$

This is confirmed on figure 10 where we see that the QBE is more accurate for small $V$ and the accuracy improves when $\omega_0$ increases.

## C    Appendix : Influence of the system-bath coupling

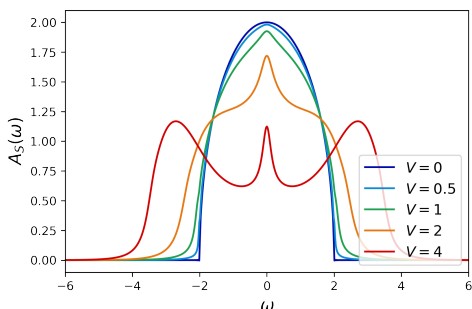

Figure 11: Spectral density of the system in equilibrium with the $\mathrm{SYK}_4$ bath with $n = 3$ coupling.

In this appendix we compare the effect of the $\mathrm{SYK}_4$ bath with two different system bath couplings, the one used in the main text and

$$H_{SB} = i \sum_{i=1}^{N} \sum_{a,b,c=1}^{M} V_{iabc}\,\chi_i \psi_a \psi_b \psi_c \tag{75}$$

We will denote these two coupling $n = 1$ and $n = 3$, where $n$ is the number of $\psi$ fermions involved.

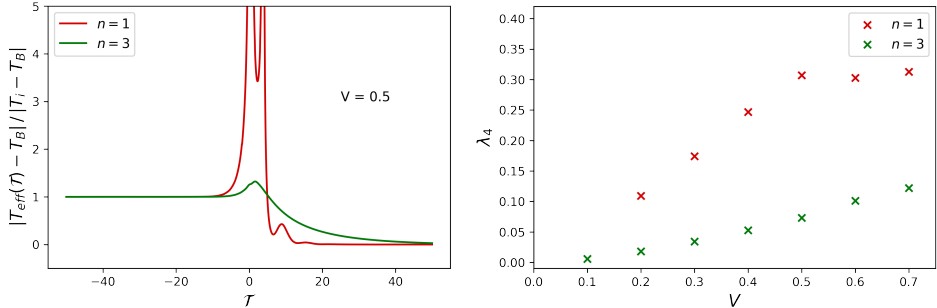

Figure 12: *Left panel*: effective temperature of the system $T_{eff}(\mathcal{T})$ for the SYK$_4$ bath with the $n = 1$ and $n = 3$ couplings. *Right panel*: long time decay rate with the $n = 1$ and $n = 3$ couplings.

With respect to the SYK$_2$ fixed point, the $n = 3$ coupling is an irrelevant perturbation. In figure 11 we show the equilibrium spectral density of the system for various values of $V$.

In figure 12 we compare the dynamics of the effective temperature with the two couplings. We see that the system heats up less and is slower to get to thermal equilibrium. In the right panel we show the long time exponential decay rate, which is smaller with the $n = 3$ coupling.

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
