# Peer review of "Are fast scramblers good thermal baths?"

_SciPost Physics_

## Round 1 · Referee Report · Anonymous · 2022-5-19

Strengths

1. The manuscript addresses the important problem of the thermalization capabilities of a non-interacting and strongly interacting baths.

2. It contains a number of analytical findings carefully supported by numerical results.

Report

I deem the paper suited for publication in SciPost Physics after some minor issues and clarifications are addressed.

Requested changes

1. Show the adequacy of the fits to Eq.(37) of the distribution function during the times immediately after the system-reservoir couplings is turned on and comment on the suitability of defining an effective temperature in these timescales.

2. Add a comment on the zero-temperature limit.

3. Comment on the possibility of the thermalization rate \lambda being obtained within linear response theory around the equilibrium state.

4. Correct a typo in Eq.(44)

---

## Round 1 · Referee Report · Anonymous · 2022-5-30

Strengths

1. Very clearly written with well-motivated questions.
2. Detailed numerical calculations supported by semi-analytic calculations.
3. Important insights into the effects of strongly interacting baths, as opposed to non-interacting baths usually used for open quantum systems, on the thermalization of a system that does not otherwise thermalize.

Weaknesses

1. No discussions on how the insights obtained using the SYK model as a bath will be useful for more realistic baths that can be realized in actual experiments.
2. The manuscript certainly satisfies several of the general acceptance criteria listed by SciPost, but it is not clear which of the expectations --i.e. detail a groundbreaking theoretical discovery, present a breakthrough .., open a new pathway .., etc., the manuscript fulfills.

Report

I will recommend publications if the authors could strengthen the introduction and more clearly state how their work goes much beyond many previous works on thermalization in SYK and related models. The authors should also add some discussions on how the work can be relevant to the more general situations, beyond only SYK-related physics, e.g. for experimentally realizable systems coupled to more realistic baths.

Requested changes

1. I think Eq.(4) has a typo; \Sigma^R(\mathcal{T},\omega) in the last term should be replaced by \Sigma^A(\mathcal{T},\omega).
2. The final temperature combined system is T_B, but in Section 5, the authors suddenly use \beta_f. Also \mathcal{T}_0 is used for the initial time in this section without properly defining it.
3. The word "equilibrate" to describe the non-thermal steady state of isolated SYK_2 is confusing. It is probably better to use "steady state" instead.

---

## Editorial Decision

awaiting_resubmission